# Explaining the Validity of the ASVAB for Job-Relevant Multitasking Performance: The Role of Placekeeping Ability

**DOI:** 10.3390/jintelligence11120225

**Published:** 2023-12-15

**Authors:** David Z. Hambrick, Alexander P. Burgoyne, Erik M. Altmann, Tyler J. Matteson

**Affiliations:** 1Department of Psychology, Michigan State University, East Lansing, MI 48824, USA; ema@msu.edu; 2School of Psychology, Georgia Institute of Technology, Atlanta, GA 30327, USA; 3Department of Psychology, Stanford University, Stanford, CA 94305, USA; tylerjm@stanford.edu

**Keywords:** ASVAB, cognitive ability, job performance, individual differences, placekeeping ability, attention control

## Abstract

Scores on the Armed Services Vocational Aptitude Battery (ASVAB) predict military job (and training) performance better than any single variable so far identified. However, it remains unclear what factors explain this predictive relationship. Here, we investigated the contributions of *fluid intelligence* (Gf) and two executive functions—*placekeeping ability* and *attention control*—to the relationship between the Armed Forces Qualification Test (AFQT) score from the ASVAB and job-relevant multitasking performance. Psychometric network analyses revealed that Gf and placekeeping ability independently contributed to and largely explained the AFQT–multitasking performance relationship. The contribution of attention control to this relationship was negligible. However, attention control did relate positively and significantly to Gf and placekeeping ability, consistent with the hypothesis that it is a cognitive “primitive” underlying the individual differences in higher-level cognition. Finally, hierarchical regression analyses revealed stronger evidence for the incremental validity of Gf and placekeeping ability in the prediction of multitasking performance than for the incremental validity of attention control. The results shed light on factors that may underlie the predictive validity of global measures of cognitive ability and suggest how the ASVAB might be augmented to improve its predictive validity.

## 1. Introduction

Decades of research have established that scores on standardized tests of cognitive ability predict job (and training) performance to a statistically and practically significant degree ([51]). One such test is the Armed Services Vocational Aptitude Battery (ASVAB). Taken annually by an estimated one million U.S. high school students, the ASVAB is the U.S. military’s primary selection and placement instrument. The ASVAB includes nine subtests (see Table 1). Criterion correlations average around 0.60 after correction for measurement error and range restriction (e.g., [37]; [56]; [44]; [47]; [34]; [23]) and are stable across different levels of job experience ([52]; [48]; [20]). Economic utility analyses indicate that using the ASVAB for military selection saves the U.S. military (and taxpayers) billions of dollars per year ([53]; [26]; [24]).

Nevertheless, it remains unclear *why* ASVAB scores predict job performance. That is, what aspects of the cognitive system does the ASVAB measure that explain its predictive validity? Presumably, part of the answer to this question is that the ASVAB captures skills and knowledge that are directly transferable to tasks in the military workplace. For example, the Paragraph Comprehension subtest measures skill in comprehending written text, which is important for work tasks such as learning how to perform a procedure from an instructional manual. As another example, the Mechanical Knowledge subtest measures knowledge of physical and mechanical principles, which is directly relevant to tasks that a mechanic or engineer performs.

In fact, evidence indicates that the ASVAB primarily measures the broad cognitive ability [12] ([12]) termed *crystallized intelligence* (Gc)—skills and knowledge acquired through experience. For example, in a confirmatory factor analysis, [49] ([49]) found that five of the nine subtests on the current ASVAB loaded with Gc tests from another test battery. However, evidence also indicates that the ASVAB measures other cognitive abilities to some degree, including *fluid intelligence* (Gf)—the ability to solve novel problems. For example, [32] ([32]) and [31] ([31]) found that scores on the Arithmetic Reasoning and Mathematics Knowledge ASVAB subtests correlated very highly with a latent factor comprising both reasoning and working memory measures. Similarly, in the above-mentioned study, Roberts et al. found that the Arithmetic Reasoning and Mathematics Knowledge subtests loaded with Gf tests from the other test battery.

There are several possible explanations for the positive relationship between ASVAB scores and Gf. One is that subtests such as Arithmetic Reasoning, Mechanical Comprehension, and Assembling Objects call upon reasoning processes, along with other cognitive abilities and knowledge/skill. Another is that ASVAB scores index variance in Gf because Gf influenced the degree to which people earlier in life, through formal and informal learning experiences, acquired the skills and knowledge the ASVAB assesses ([13]).

The ASVAB may also capture *executive functions*: domain-general cognitive abilities involved in effortfully guiding behavior toward goals, especially under non-routine circumstances ([5]). In recent years, two executive functions have been proposed that seem particularly relevant to explaining the predictive validity of the ASVAB. The first is *attention control:* the ability to maintain task-relevant information in a highly active state within working memory, especially under conditions of distraction or interference ([22]; [7]). The second is what we have termed *placekeeping ability:* the ability to perform a sequence of mental operations in a prescribed order without skipping or repeating steps ([4]). Placekeeping performance presumably involves attention control, but may, in addition, capture operations involved in a broad class of tasks with a procedural component in which keeping track of one’s place in a set of steps is important.

The ASVAB may measure attention control because, to answer a test item correctly, the test taker presumably must attend to it, and placekeeping ability, as some of the test items require the test taker to follow a procedure to solve a problem (e.g., multiplying binomials). Alternatively, just as they may for Gf, ASVAB scores may index attention control and placekeeping ability, if these influenced the extent to which participants acquired the skills and knowledge the ASVAB measures. Scores on the ASVAB could also correlate with scores on tests of executive functions such as attention control and placekeeping ability because the different instruments “sample” some of the same domain-general cognitive processes, including not only those executive functions but any number of other domain-general cognitive processes as well (see [30]).

Regardless of whether measures of placekeeping ability and attention control explain the predictive validity of the ASVAB, they may improve the prediction of job-relevant performance above and beyond the ASVAB. In other words, ASVAB scores and executive control measures may account for unique portions of variance in job-relevant performance. If so, tests of executive functions would be promising candidates to augment the ASVAB.

## 2. Present Study

Our major research question was whether Gf, placekeeping ability, and attention control would explain the relationship between AFQT score and performance in “synthetic work” tasks designed to measure job-relevant multitasking performance. The goal of the synthetic work approach, developed in the 1960s by human factors psychologists ([2]; [39]), is to create laboratory tasks that capture requirements common to a broad class of work tasks rather than to simulate a specific job. Thus, a synthetic work task does not resemble any actual work task *by design.* The most widely administered synthetic work task was developed for the U.S. Army by [19] ([19]). The participant is to manage component tasks presented in quadrants of their computer screen: a self-paced math task (upper right), a memory probe task (upper left), a visual monitoring task (lower left), and an auditory monitoring task (lower right; further details are presented below). Total points, displayed in the center of the screen, is the sum of points earned in the component tasks. The synthetic work approach helps predict performance on classes of jobs, including potential jobs of the future workplace, complementing selection instruments that measure performance on specific jobs.

We asked whether Gf, placekeeping ability, and attention control are distinct dimensions of variation in cognitive functioning that contribute to individual differences in multitasking, a performance context in which “cognitive processes involved in performing two (or more) tasks overlap in time” ([29]). The role of Gf in multitasking may be to develop a strategy (or plan) for sequencing the component tasks based on the payoffs of the component tasks and considerations about when it is advantageous to sacrifice performance in one component task to attend to another. The role of placekeeping ability may be to implement the strategy or plan, allowing the task performer to keep track of which component tasks have been performed and which remain to be performed. Finally, attention control may serve multiple functions in multitasking, such as maintaining focus on a given component task while suppressing information from other component tasks, disengaging from one component task to shift to another one, and monitoring performance.

The traditional approach to answering a research question such as ours is structural equation modeling (SEM). In SEM, *latent variables* are specified as “common causes” of individual differences in sets of *observed variables*, and then causal relations among the latent variables are estimated via path analysis. However, there is not always a strong justification for making causal assumptions concerning relations among variables. For example, attention control is a plausible cause of individual differences in the reasoning tests we used to measure Gf, as the ability to maintain information from a reasoning problem in working memory is presumably critical for being able to solve that problem. However, Gf is a plausible cause of individual differences in the tasks we used to measure attention control, given that those tasks were novel to participants, and given evidence that strategy use influences performance in even very simple cognitive tasks (e.g., [16]).

*Psychometric network analysis* offers an alternative approach to investigating relations among psychological variables that does not rest on assumptions about causality (see [50]; [6]; [38]). Each variable in the analysis is represented by a *node* (a circle).[note 1] The nodes are connected by non-directional *edges* (lines) and association strength is described with partial correlations (or some other statistical parameter), with thicker edges indicating stronger associations. All possible nondirectional associations are identified, and thus the graphical representation of the results (the *network model*) captures all possible ways that any two variables may relate to each other. In a network model of cognitive ability variables where values for links are partial correlations, the link between two variables represents the association between those variables controlling for the *g* factor in the data ([38]).

Here, to answer our major research question, we generated and compared two network models in which the nodes were composite variables (represented by circles) created by averaging the observed variables corresponding to the different factors. In the *bivariate model*, association strength was indicated by bivariate correlations (Pearson’s *r*s). In the *partial model*, association strength was indicated by partial correlations (*pr*s), each reflecting the association between two variables statistically controlling for the other variables in the model. A finding that the bivariate correlation between AFQT and multitasking performance was significant would be consistent with existing research and practice. A finding that the corresponding partial correlation was also significant would indicate that variability in AFQT directly explains variability in multitasking performance after partialling out the variance shared with other measures collected in this study. In contrast, a finding that the partial correlation was nonsignificant may indicate the relationship between AFQT and multitasking performance was explained by one or more of the other predictor variables (Gf, attention control, and placekeeping ability).

An additional goal of this study was to evaluate the incremental validity of Gf, placekeeping ability, and attention control in the prediction of multitasking performance. In an influential series of studies, Ree, Carretta, and colleagues showed that variance specific to the ASVAB subtests (“s” factors) added little to the prediction of job (and training) performance over variance common to the subtests (e.g., [47]; [44]). Furthermore, in factor-analytic studies, correlations between factors from the ASVAB and those representing constructs from cognitive psychology such as working memory capacity have been found to be very high ([31]), although less than unity ([28]; [1]). The view that has emerged from such evidence is that predicting job performance is “not much more than *g*” ([47]) as measured by the ASVAB. Recently, [45] ([45]) concluded that predicting job performance is “*still* not much more than *g*” (p. 1, emphasis added).

This view implies that the ASVAB measures all aspects of the cognitive system that may bear on job performance. Yet, the predictive validity of the ASVAB is far from perfect. For example, if the validity of the ASVAB across all jobs is 0.60, then well over half of the variance in job performance (64%) is explained by factors other than those measured by the ASVAB (i.e., 100 − 0.60^2^ × 100 = 64). At least some of this unexplained variance may be explained by Gf, placekeeping ability, and attention control. These factors have all been found to correlate strongly with multitasking performance as measured with synthetic work tasks ([10]; [21]; [43]; [36]), but the present study is the first to measure all three factors and multitasking performance in the same sample of participants.

We took two approaches to testing the incremental validity of Gf, placekeeping ability, and attention control for multitasking. As a weaker test, we performed hierarchical regression analyses to evaluate the extent to which Gf, placekeeping ability, and attention control each predicted multitasking performance over AFQT. As a stronger test, we evaluated the extent to which each factor improved the prediction of multitasking performance over AFQT and the other factors we measured.

## 3. Method

### 3.1. Participants

Participants were undergraduate students from a large midwestern (U.S.) university recruited through the Department of Psychology subject pool and received credit toward a course requirement for volunteering. The study was approved by the university’s Institutional Review Board, and all participants gave informed consent at the beginning of the study.

Participants ranged in age from 18 to 26 years (*M* = 19.0, *SD* = 1.1); 70.5% reported they were female, 26.9% reported they were male, and 2.6% did not report gender. The breakdown of self-reported ethnic/racial group was Black (11.4%), White (72.0%), Asian (10.3%), Other (3.7%), and no response (2.6%).

With respect to college admissions scores, the mean ACT score (*n* = 112) was 26 (*SD* = 4, Range = 16–35), compared to a mean of 21 for all test takers (*SD* = 6, Range = 1 to 36), and the mean SAT score (*n* = 175) was 1195 (*SD* = 134, Range = 880–1550), compared to a mean of 1059 for all test takers (SD = 210, Range = 400–1600; [40]). Thus, relative to all test takers, the mean ACT score was 0.9 *SD*s higher and the *SD* was 33% smaller, and the mean SAT score was 0.7 *SD*s higher and the *SD* was 37% smaller. Furthermore, on average, the four ASVAB subtest means were 0.5 *SD*s above means for a large sample of applicants (*N* = 8808) for military service ([46]). In sum, the sample was higher on average than the general population in cognitive ability, and although there was some degree of range restriction, it was not severe.

Of the 270 participants who completed Session 1, 29 did not return for Session 2 (leaving *n* = 241), another 26 did not return for Session 3 (leaving *n* = 215), and another 57 did not return for Session 4 (leaving *n* = 158). It appears attrition was random with respect to performance on the predictor and criterion tasks. The point-biserial correlations of the performance measures from each session with a binary variable indicating whether a participant showed up for the next session (0 = no, 1 = yes) were all near zero and nonsignificant (mean *r*_pb_ = 0.04). A likely reason for the markedly higher attrition from Session 3 to 4 is that after three sessions participants had earned enough experiment credits to fulfill the research participation assignment in their course.

### 3.2. Procedure and Materials

This study took place in four sessions, each lasting 1 to 1.5 h. Participants were tested in groups of up to 7. As is standard in individual difference research (e.g., [36]), the measures were collected in a fixed order to avoid participant × order interactions. Table 2 lists the tests for each session, in the order administered.[note 2]

*Armed Services Vocational Aptitude Battery (ASVAB).* Participants completed four subtests from a practice ASVAB. The four subtests were those comprising the Armed Forces Qualification Test (AFQT), which is used to determine whether an individual is eligible to enlist in the military.[note 3] Each test consisted of 4-alternative multiple-choice items. The number of items per test and time limits were the same as in the actual ASVAB. The subtests were presented to participants in paper-and-pencil format in binders, but participants entered their answers for each subtest in a Qualtrics form. Keeping time with a digital stopwatch, an experimenter instructed participants when to begin working on each subtest and when to stop.

The four subtests comprising the AFQT were Arithmetic Reasoning, Word Knowledge, Paragraph Comprehension, and Mathematics Knowledge (see Table 1). Arithmetic Reasoning consisted of 30 word problems, and the time limit was 36 min. Word Knowledge consisted of 35 synonym questions, and the time limit was 11 min. Paragraph Comprehension consisted of 15 questions, of which the first 12 each referred to a separate paragraph and the last three referred to the same paragraph; the time limit was 13 min. Mathematics Knowledge consisted of 25 questions to assess knowledge of basic mathematic concepts, and the time limit was 24 min. For each subtest, the score was the number correct.

*Attention control (AC).* There were five attention control tasks (see additional details about the tasks in [36], and [17]).[note 4] In a trial of Sustained Attention to Cue, the participant saw a fixation cross for 2 or 3 s (quasi-randomly determined, half the trials for each duration), after which a 300 ms tone was played, followed by a white circle appearing in a randomly determined location, not at the center of the screen. After the circle onset, it shrank for 1.5 s until it reached a fixed size, after which there was a variable wait time of 2, 4, 8, or 12 s (quasi-randomly determined, equal numbers of trials for each duration). After the wait time, a distractor in the form of a white asterisk blinked on and off at the center of the screen for 400 ms. After the blinking asterisk, a 3 × 3 array of letters appeared at the location of the circle cue. The central letter in the array (one of B, D, P, and R) was the target and appeared in dark gray font. The other letters in the array (two each of B, D, P, and R) appeared in black font. After 125 ms the target was masked with a # character for 1 s, after which the array offset and four response boxes appeared on the screen, each containing one of the four possible targets. The participant was to choose the box with the target letter. The measure of performance was accuracy (the proportion of correct trials).

In a trial of Antisaccade, the participant saw a fixation cross for a randomly determined amount of time between 2000 and 3000 ms, after which an alerting tone was played for 300 ms. A distractor (a flashing # character) then appeared on either the left or right side of the screen for 300 ms and was followed immediately by a target letter (Q or O) for 100 ms on the side of the screen opposite to where the distractor appeared. Finally, a mask (two # characters) appeared for 500 ms at the location of the target and the distractor. The participant was to identify the target (Q or O) by pressing the corresponding key. The measure of performance was accuracy (the proportion of 72 trials that were correct).

In a trial of Flanker Deadline, the participant saw five arrows and was to indicate which direction the middle arrow was pointing. A response deadline, announced by a loud beep, limited how long the participant was allowed to respond. Each of the 18 blocks included 12 congruent and 6 incongruent trials, presented in random order with a randomized inter-stimulus interval between 400 and 700 ms. The starting deadline was 1050 ms. Between blocks, the deadline decreased if the participant was correct on at least 15 of 18 trials (by 90 ms for Blocks 1–6 and 30 ms for Blocks 7–18) and increased if they were not (by 270 ms for Blocks 1–6 and 90 ms for Blocks 7–18). The measure of attention control was what the response deadline would be for a 19th block, multiplied by −1 so that higher values would indicate better performance.

In a trial of Selective Visual Arrays, the participant saw a display composed of blue and red rectangles in different orientations on a light gray background, with 5 or 7 rectangles of each color (10 or 14 rectangles in total). Before each trial, the participant was cued to attend to either the red or blue rectangles. After a 900 ms delay, the target array appeared for 250 ms, after which an array with only the rectangles of the target color appeared. One rectangle was circled, and the participant was to respond whether it was in the same orientation as in the initial array. The rectangle was in the same orientation on 50% of trials, and there were 48 trials per array size, for 96 trials total. The measure of attention control was the *k* measure of capacity [N × (% hits + % correct rejections − 1), where *N* = display size], averaged across array sizes.

In a trial of the Stroop Deadline, the participant saw a color word in the congruent or incongruent color. The beginning deadline was 1230 ms; otherwise, task parameters were the same as in Flanker. The measure of attention control was what the response deadline would be for a 19th block, multiplied by −1 so higher values would indicate better performance.

*Fluid intelligence (Gf).* There were three Gf tests. In Raven’s Advanced Progressive Matrices ([42]; [43]), each item consisted of a 3 × 3 array of geometric patterns, with the pattern in the lower right missing. Eight alternatives appeared below the array, and the participant’s task was to click on the one that fit logically as the missing pattern. We used the 18 odd items from Raven’s, and the score was the proportion correct. The time limit was 10 min.

In Letter Sets ([18]; [43]), each item consisted of five sets of letters in a row, with four letters per set (e.g., FQHI DFHJ YLMH ERHT VNHQ). The participant was to click on the set that did not fit logically with the other sets. There were 20 items, and the score was the proportion correct. The time limit was 10 min.

In Number Series ([54]; [43]), each item consisted of a series of numbers, followed by a blank space (e.g., 1 3 6 10 __). The participant was to infer the rule governing the changes in the digits and to select the alternative that fit logically in the blank. There were 15 items, and the score was the proportion correct. The time limit was 5 min.

*Multitasking performance (MP).* Three synthetic work tasks were used to assess multitasking performance (see Appendix F for screenshots). In the Foster Task (developed by Jeffrey L. Foster, Macquarie University; see [36]), four subtasks are displayed in quadrants of the screen. In Telling Time, the participant saw a clock and was to click on one of four buttons showing the time displayed. In Visual Monitoring, the participant was to click on a disk as quickly as possible after it began spinning. In Word Recall, a word appeared in green and then disappeared. A word then appeared in red and the participant was to click a YES button if it was the same word or a NO button if it was not. In Math, the participant was to solve two-term addition problems. In each subtask, participants were awarded 100 points for correct responses and docked 100 points for incorrect responses. Participants completed a 5 min session of the Foster task; the measure of multitasking performance was the average score across the subtasks.

In Control Tower ([43]), participants were to perform a primary task while concurrently performing secondary tasks. In the primary task, the participant was shown two arrays of digits, letters, and symbols, which were presented in adjacent rectangles. The participant was to search through the left array and respond to items in the right array, according to the following rules: for a digit, click on the same digit; for a letter, click on the preceding letter in the alphabet; and for a symbol, click on a corresponding symbol, clicking on a CODE BOOK button on the right side of the screen to display the symbol pairings if necessary. While performing this task, the participant was to respond to interruptions from four secondary tasks. In the Radar task, the participant was to monitor a circular display in the bottom left of the screen, clicking an INSIDE button for blips appearing inside the circle and an OUTSIDE button for those appearing outside of it. In the Airplane task, the participant was to respond to landing requests presented through headphones by clicking on a RUNWAY button to view runway availability and then clicking on a CLEAR FOR LANDING button or DENY LANDING button based on this information. In the Color task, one of three colors was flashed on the screen and the participant was to click on the appropriate “error” button (ERROR 1, ERROR 2, or ERROR 3), clicking on a PROTOCOL button on the right side of the screen to view the color-error pairings. Finally, in the Problem Solving task, the participant heard trivia questions through the headphones and was to click on one of three possible answers at the bottom of the screen. Participants completed a 10 min session. The measure of multitasking performance was the average of *z* scores for the primary task and distractor task scores.

In SynWin ([19]), the participant was to manage four subtasks, each presented in a different quadrant of the screen with the total score displayed in a box in the center of the screen. In Arithmetic (upper right), the participant was to add numbers using the + and − buttons and then click “Done” to register the answer. Ten points were awarded for correct responses and 10 points were deducted for incorrect responses. In Memory Search (upper left), a set of seven letters was displayed at the beginning of the session and then disappeared; probe letters were then displayed at a regular interval, and the participant’s task was to judge whether each was from the set by clicking on a Yes or No button. Ten points were awarded for correct responses and 10 points were deducted for incorrect responses or failures to respond. In Visual Monitoring (lower left), a line representing a needle moved from right to left on a fuel gauge, and the participant was to click anywhere on the gauge before the needle reached the end of the red region. Ten points were awarded for a reset occurring when the needle was in the red region and fewer points for resets occurring earlier. Finally, in Auditory Monitoring (bottom right), the participant was to respond to a high-pitched tone by clicking on an “Alert” button, while ignoring low-pitched tones. Ten points were awarded for correct responses and 10 points were deducted for incorrect responses or failures to respond. There was a 1 min practice block, followed by four 5 min test blocks. The measure of multitasking performance was the total score.

*Placekeeping ability (PA).* There were two placekeeping tasks, each of which involved performing a “looping” procedure (see Appendix F for screenshots of the materials, and [3] and [9] for further details). In each trial of UNRAVEL, the participant applied a 2-alternative forced-choice (2AFC) decision rule to a randomly generated stimulus with features including a letter, a digit, and several kinds of formatting. There were seven rules, each linked mnemonically to a letter of the word UNRAVEL as follows: (1) character underlined or in italics, (2) letter near to or far from the beginning of the alphabet, (3) character red or yellow, (4) character above or below an outline box, (5) letter a vowel or consonant, (6) digit even or odd, and (7) digit less or more than 5. For each rule, the two possible responses were the first letters of the two options (e.g., in Step 1, *u* if the letter is underlined or *i* if it is italicized). (Note that the first responses spell UNRAVEL.) The next trial began immediately after the response. On the next trial, a new stimulus appeared, and the participant was to shift to the next rule in the UNRAVEL sequence, with U following L (creating the “loop”). A placekeeping error occurred if the participant applied an incorrect rule relative to the previous trial (e.g., the R rule following the U rule). The stimulus contained no information about the participant’s location in the rule sequence, so the participant had to remember this information. For help remembering the rule sequence and responses, the participant could press a key sequence to display a help screen.

Performance was periodically interrupted by a typing task that replaced the placekeeping stimulus on the display. On each trial of the typing task, the participant typed a 14-letter “code” into a gray box. The code was made up of the candidate UNRAVEL responses presented in random order. The participant could backspace to correct mistakes and pressed the Enter/Return key to enter the code when they deemed it complete. If the entered code was incorrect, the box was cleared and the participant had to re-enter the code. There were two trials of the typing task per interruption. After the second code was entered, the participant was to resume the UNRAVEL sequence with the rule following the rule applied on the last trial before the interruption. Participants completed two blocks of UNRAVEL, with a block comprising 11 runs of 2AFC trials separated by 10 interruptions.

In each trial of Letterwheel, the participant saw a “wheel” of nine letters and was to alphabetize a set of three letters located at contiguous spatial positions. The first two letters appeared in a box at the center of the wheel as they were typed, and the participant could backspace to correct mistakes. When the third letter was typed the next trial began immediately. On the next trial, the locations of the nine letters were randomized, and the set of positions to alphabetize shifted one position clockwise on the wheel. The set of positions continued to shift one position per trial on subsequent trials, circling the wheel repeatedly (creating the “loop”). A placekeeping error occurred when the participant typed letters from an incorrect set of positions and was scored with respect to the previous trial. For help remembering the alphabet the participant could press a key sequence to display it.

Performance was periodically interrupted by a counting task that replaced the letter wheel on the display. On each trial, 7, 8, or 9 asterisks were randomly distributed on the display within the same region occupied by the letter wheel, and the participant responded by pressing the key corresponding to the number of asterisks. If the response was incorrect, the screen flashed and the participant repeated the trial with the same asterisk display. There were five counting trials per interruption, and after the fifth, the participant was to resume alphabetizing at the correct set of positions on the wheel, namely one position clockwise from the alphabetizing trial immediately before the interruption. Participants completed two blocks of Letterwheel, with a block comprising nine runs of alphabetizing trials separated by eight interruptions.

For each placekeeping task, we created a composite performance variable capturing all aspects of performance in the task (see the Appendix of [10]). This composite was the average of *z* scores for baseline sequence error rate, post-interruption sequence error rate, nonsequence error rate, baseline response time on correct trials, post-interruption response time on correct trials, time to complete the training phase before the test phase, and response time for interruptions, multiplied by −1 so that higher values would indicate better performance.

### 3.3. Data Preparation

We prepared the data for analysis in four steps. First, we deleted values of 0 or less (points) on multitasking variables, 0 (number of problems correct) on ASVAB and Gf variables, 0 (proportion of trials correct) on attention control variables, and 1 (proportion of trials incorrect) on placekeeping variables. Our assumption was that values not meeting these criteria reflected equipment error, or participants making no effort. Second, we converted the variables to *z* scores and screened for univariate outliers, defined as a *z* score greater or less than 3.5 (i.e., > or <3.5 standard deviations from the total sample mean). Third, we replaced any outlier with a value of *z* = 3.5 or −3.5, as appropriate. Finally, we used the expectation maximization (EM) procedure in SPSS 27 to replace values missing due to attrition or technical or experimenter error (18.4% of the data) and values missing due to the above deletions (0.6% of the data). Frequencies of outliers and missing and replaced values are reported in Appendix A.

Our final data set included 270 participants, but to evaluate whether estimating missing data materially affected conclusions, we performed all major analyses reported in the *Results* section using the subsample of participants who completed all four sessions (*n* = 158). The results of these analyses are summarized below and presented in Appendix C. Data files are available on OSF (https://osf.io/z4gf3/?view_only=0e79562fe39f4b0e83294f38630c5f0d).

### 3.4. Composite Variables

Using the cleaned data set, we computed the AFQT score for each participant by converting the subtest scores into *z* scores, and then, following the practice used in military applications, applying the formula: 2(Word Knowledge *z* + Paragraph Comprehension *z*) + Mathematics Knowledge *z* + Arithmetic Reasoning *z*. For the other factors, based on classifications of the tasks in the extant literature, we created composite variables by averaging *z* scores of the observed measures, as follows: Gf (Raven’s Matrices, Letter Sets, Number Series), Attention Control (Antisaccade, Sustained Attention to Cue, Flanker Deadline, Stroop Deadline, Selective Visual Arrays), Placekeeping Ability (UNRAVEL composite, Letterwheel composite), and Multitasking Performance (Foster, Control Tower composite, Synthetic Work). See Appendix A for descriptive statistics and a full correlation matrix.

We also performed a psychometric network analysis on the individual observed variables (see Appendix B). The results show mixed evidence for the coherence of the factors we adopted from the extant literature. As expected, there was evidence that the ASVAB subtests used to compute the AFQT score measure two factors: mathematical ability (Arithmetic Reasoning and Mathematics Knowledge, *pr* = 0.45) and verbal ability (Word Knowledge and Paragraph Comprehension, *pr* = 0.33). There also was evidence for a placekeeping ability factor, as there was a significant link between UNRAVEL and Letterwheel (*pr* = 0.35), consistent with previous findings ([3]). By contrast, although there was a significant link between two of the three Gf variables (Raven’s and Letter Sets, *pr* = 0.25), the third Gf variable (Number Series) sat apart in the network model. Furthermore, although some of the links between attention control variables were significant (e.g., Selective Visual Arrays and Sustained Attention to Cue, *pr* = 0.27), others were not (e.g., Flanker Deadline and Sustained Attention to Cue, *pr* = −0.01). This result suggests that some cognitive constructs may be less psychologically coherent than others, a point to which we return in the Discussion.

### 3.5. Power Analyses

For the psychometric network analyses, we conducted post-hoc power analyses (in SPSS 27) for bivariate correlations and partial correlations, assuming that three variables were partialled out in the latter. With alpha set at 0.05, a sample of 270 provided power of 0.80 for *r* = *pr* = 0.17, 0.90 for *r* = *pr* = 0.20, and 0.99 for *r* = *pr* = 0.25. For the hierarchical regression analyses, we conducted post-hoc power analyses for increment in variance explained (Δ*R*^2^), assuming four variables in the model with one test variable, as in our more stringent test of incremental validity. With alpha set at 0.05, a sample size of 270 provided a power of 0.80 for ∆*R*^2^ = 0.029, 0.90 for ∆*R*^2^ = 0.038, and 0.99 for ∆*R*^2^ = 0.064.

## 4. Results

### 4.1. Psychometric Network Analyses

Our primary research question was whether Gf, Placekeeping Ability, and Attention Control would explain the relationship between AFQT and Multitasking Performance. To answer this question, we performed psychometric network analyses to generate two network models, presented in Figure 1. We used the Network modules in JASP Version 0.17.2; all analysis scripts are available upon request. In the first model (left panel), the values are bivariate correlations (*r*s). In the second model (right panel), the values are partial correlations (*pr*s), each reflecting the association between two variables statistically controlling for their associations with the other three variables in the model. (The weight matrices for the models are presented in Table 3.) A finding that the bivariate correlation between two variables (e.g., AFQT and Multitasking Performance) is substantially larger than the partial correlation between those variables would be evidence that one or more of the other variables in the model explained the former correlation.

The AFQT–Multitasking Performance bivariate correlation (*r* = 0.49) was significant whereas the corresponding partial correlation was nonsignificant (*pr* = 0.09), indicating that one or more of the other variables in the model explained the AFQT–Multitasking Performance bivariate correlation. The bivariate correlations between AFQT and Gf (*r* = 0.54), Placekeeping Ability (*r* = 0.53), and Attention Control (*r* = 0.40) were all significant, whereas the corresponding partial correlations were significant for Gf (*pr* = 0.25) and Placekeeping Ability (*pr* = 0.20) but not Attention Control (*pr* = 0.04). This pattern indicates that the AFQT captured variance related to Gf and Placekeeping Ability but not Attention Control. The bivariate correlations between Multitasking Performance and Gf (*r* = 0.63), Placekeeping Ability (*r* = 0.69), and Attention Control (*r* = 0.51) were all significant, whereas the corresponding partial correlations were significant for Gf (*pr* = 0.29) and Placekeeping Ability (*pr* = 0.42) but not Attention Control (*pr* = 0.09). Taken together, these findings suggest that AFQT score predicted Multitasking Performance in the bivariate model because at least some of the variability in AFQT scores was related to Gf and Placekeeping Ability.

To evaluate the strength of the evidence for our conclusion, we performed a Bayesian analysis on the partial correlation network (using the JASP Version 0.17.2 Bayesian Network module, Gaussian Graphical Model estimator). The Bayes Factor (BF_10_) is the ratio of the likelihood of the data given the alternative hypothesis, P(D|H_1_), to the likelihood of the data given the null hypothesis, P(D|H_0_). Accordingly, a BF_10_ > 1 supports the alternative hypothesis, whereas a BF_10_ < 1 supports the null hypothesis. For example, BF_10_ = 5 indicates that the data are 5 times more likely to have occurred under the alternative hypothesis than under the null hypothesis, whereas BF_10_ = 0.20 indicates that the data are 5 times more likely to have occurred under the null hypothesis than under the alternative hypothesis (i.e., 1/0.20 = 5). Following convention (e.g., [35]), we characterized the strength of evidence for one hypothesis over the other as “weak” for a ratio of 1 to 3, “moderate” for a ratio of 3 to 10, “strong” for a ratio of 10 to 30, and “very strong” for a ratio greater than 30.

The results are shown in Table 4 and Figure 2. For each edge (path), the alternative hypothesis is that the partial correlation has an absolute magnitude greater than zero (H_1_: *pr* > 0), whereas the null hypothesis is that the partial correlation is zero (H_0_: *pr* = 0). The edge inclusion criterion in Figure 2 was a BF_10_ > 3. There was very strong evidence for H_1_ for the edges between AFQT and both Gf and Placekeeping Ability (BF_10_s > 100), between Gf and both Attention Control and Multitasking Performance (BF_10_s > 100), and between Attention Control and Placekeeping Ability and Placekeeping Ability and Multitasking Performance (BF_10_s > 100). Furthermore, there was moderate or better evidence for H_0_ for the edges between AFQT and both Attention Control (BF_10_ = 0.06) and Multitasking Performance (BF_10_ = 0.12) and between Attention Control and Multitasking Performance (BF_10_ = 0.15), and weak evidence for H_0_ for the path between Gf and Placekeeping Ability (BF_10_ = 0.43).

Taken together, these results are compelling evidence that AFQT predicted Multitasking Performance in our sample because it indexed Gf and Placekeeping Ability, and that Attention Control played no role in the relationship between AFQT and Multitasking Performance. We repeated the psychometric network analyses using the subsample of participants who completed all sessions; the conclusions were the same as with the full sample (see Appendix C, Figure A3).

### 4.2. Hierarchical Regression Analyses

We performed hierarchical regression analyses to test for the incremental validity of Gf, Placekeeping Ability, and Attention Control in the prediction of Multitasking Performance over and above the AFQT. The results are presented in Table 5, Table 6 and Table 7, including the change in variance explained (∆*R*^2^) for the steps of each model, along with the standardized regression coefficient (β) for each predictor variable entered in those steps, reflecting the predicted increase in Multitasking Performance in *SD* units for every 1 *SD* increase in the predictor, controlling for the other predictor(s) in the model.

In the first set of models (Table 5), we tested the incremental validity of each cognitive factor over AFQT. AFQT explained 24.1% of the variance (Δ*F* = 85.17, *p* < .001, β = 0.49). Gf explained 19.3% of the variance over AFQT (Model 1A, Δ*F* = 91.06, *p* < .001, β = 0.52). Placekeeping Ability explained 25.7% of the variance over AFQT (Model 1B, Δ*F* = 136.62, *p* < .001, β = 0.60). Attention Control explained 11.9% of the variance over AFQT (Model 1C, Δ*F* = 49.54, *p* < .001, β = 0.38).

In the second set of models (Table 6), we tested the incremental validity of each cognitive factor over AFQT and each other cognitive factor. Gf explained 5.5% of the variance over AFQT and Placekeeping Ability (Model 2A, Δ*F* = 32.46, *p* < .001, β = 0.31), and 10.0% of the variance over AFQT and Attention Control (Model 2B, Δ*F* = 49.42, *p* < .001, β = 0.42). Placekeeping Ability explained 11.8% of the variance over AFQT and Gf (Model 2C, Δ*F* = 70.43, *p* < .001, β = 0.46), and 15.6% of the variance over AFQT and Attention Control (Model 2D, Δ*F* = 85.87, *p* < .001, β = 0.52). Attention Control explained 2.6% of the variance over AFQT and Gf (Model 2E, Δ*F* = 12.84, *p* < .001, β = 0.20), and 1.8% of the variance over AFQT and Placekeeping Ability (Model 2F, Δ*F* = 9.95, *p* < .001, β = 0.16).

In the third set of models (Table 7), we tested the incremental validity of each cognitive factor over AFQT and both other cognitive factors. Gf explained 4.0% of the variance over AFQT, Placekeeping Ability, and Attention Control (Model 3A, Δ*F* = 24.17, *p* < .001, β = 0.28). Placekeeping Ability explained 9.6% of the variance over AFQT, Gf, and Attention Control (Model 3B, Δ*F* = 57.59, *p* < .001, β = 0.43). Attention Control explained a nonsignificant 0.4% of the variance over AFQT, Gf, and Placekeeping Ability (Model 3C, Δ*F* = 2.36, *p* = .125, β = 0.08).

To sum up, Gf, Placekeeping Ability, and Attention Control each improved the prediction of Multitasking Performance over AFQT. Moreover, each factor improved the prediction of Multitasking Performance over AFQT and each other factor, although the increments in variance explained were notably higher for Gf and Placekeeping Ability than for Attention Control. Finally, Gf and Placekeeping Ability each improved the prediction of Multitasking Performance over AFQT and both other factors, whereas Attention Control did not. Thus, overall, evidence for incremental validity was stronger for Gf and Placekeeping Ability than for Attention Control. 

In Appendix C we repeat the hierarchical regression analyses using the subsample of participants who completed all sessions; the conclusions were the same as with the full sample. In Appendix E we report simultaneous regressions of Multitasking Performance on all four predictors individually and in combination, to support an assessment of the amount of criterion variance explained as a function of number of predictors.

## 5. Discussion

The development of reliable and valid tests of cognitive ability for the prediction of job performance (among other uses) is arguably one of applied psychology’s greatest achievements. The ASVAB, and more specifically the AFQT score, is a compelling case in point. However, it is unclear what cognitive factors explain the predictive validity of the ASVAB. In this study, we investigated the contributions of three cognitive abilities—Gf, placekeeping ability, and attention control—to the relationship between the AFQT score and job-relevant multitasking performance.

We compared two psychometric network analyses of relations among composite variables representing the predictor and criterion (see Figure 1). The correlation between AFQT and multitasking performance was strongly positive, but the corresponding partial correlation was near zero. Moreover, although the correlations of both AFQT and multitasking performance with the other predictor variables were strongly positive, the corresponding partial correlations were significant for Gf and placekeeping ability but not attention control. The results are evidence that Gf and placekeeping ability contributed independently to the relationship between AFQT and multitasking performance, whereas attention control did not.

These findings are consistent with the possibility that Gf and placekeeping ability are causal factors underlying individual differences in multitasking. Gf measures the ability to develop solutions to novel problems, and placekeeping measures the ability to maintain one’s place in a sequence of mental operations without omissions or repetitions. Thus, one interpretation of the independent contributions of the two factors to multitasking performance is that Gf measures success at developing effective multitasking strategies, and placekeeping ability measures success at executing those strategies once they are developed. The significant partial correlations of attention control with Gf and placekeeping suggest that attention is a primitive cognitive operation underlying both ([7]).

Following up on a study by [36] ([36]), we evaluated the incremental validity of Gf, placekeeping ability, and attention control in the prediction of multitasking performance. Novel in our study were the additions of Gf and placekeeping ability. Each of Gf, placekeeping ability, and attention control improved the prediction of multitasking performance over AFQT, but only Gf and placekeeping ability did so over all other predictor variables (i.e., AFQT and, respectively, placekeeping ability and Gf). These results indicate that tests of Gf and placekeeping ability are perhaps the most promising candidates for augmenting the ASVAB to predict military job performance, at least in military occupation specialties that place high demands on multitasking. Appendix E presents additional information about how the ASVAB might be augmented, in the form of *R*^2^ values for each predictor variable alone and in all possible combinations.

The effect sizes for attention control in the present study were smaller than those in [36] ([36]). In particular, Martin et al. found that three attention control measures (Antisaccade, Sustained Attention-to-Cue, and Selective Visual Arrays) explained 6.4% of the variance in multitasking performance over ASVAB and Gf. By contrast, in the present study, the attention control composite explained 2.6% of the variance over ASVAB and Gf (see Table 6, Model 2E). This difference does not appear to be due to our measures of attention control having poor psychometric properties. Reliability estimates for our measures of attention control, reported in Table A2, were similar in magnitude to those reported by [36] ([36]). Furthermore, indicative of construct validity, the five measures of attention control correlated moderately and positively with each other, with an average correlation of 0.40 (range = 0.27–0.56; see Table A2). The average correlation among the three measures of attention control that Martin et al. used in their main analyses was very similar (avg. *r* = 0.43; range = 0.33–0.47). The weaker effect sizes in our study may instead reflect a greater range of cognitive ability in the Martin et al. sample than in ours, given that their sample included both undergraduate students and community-recruited non-students, whereas ours included only undergraduate students. In fact, there was greater variability in some of the attention control measures in Martin et al.’s study than in our study (e.g., Antisaccade *SD*s of 0.15 and 0.11, respectively). Finally, the difference could also be due to sampling error, as the sample sizes in Martin et al.’s study (*N* = 171) and this study (*N* = 270) were typical for this area of research though still somewhat small.

Concerning our finding of stronger incremental validity for placekeeping ability than for attention control in the prediction of multitasking performance, we offer two explanations that are not mutually exclusive. First, placekeeping ability constitutes a complex suite of cognitive processes, whereas attention control represents a set of more basic cognitive operations. Thus, placekeeping may have been relatively closer to, and thus accounted for more variance in, the outcome variable. Second, our composite measure of performance on each placekeeping task (UNRAVEL and Letterwheel) comprised seven separate measures that independently and exhaustively captured all aspects of performance on that task (as described in the Appendix of [10]). By contrast, for attention control, there was only a single measure of performance for each task. Thus, placekeeping ability may have explained more variance in the criterion variable because our placekeeping measures captured more aspects of performance. One might argue that this difference in the scope of the measures made for an unfair comparison of the two constructs in terms of incremental validity. However, from both a theoretical and a practical perspective, we could think of no particular reason to exclude any of the performance measures from the placekeeping tasks. We further note that it would likely be profitable to take this “whole task” approach in the measurement of attention control, among other executive functions. For example, a composite measure of performance on the antisaccade task might include the time to proceed through instructions that require performing practice trials correctly. A study comparing incremental validity of placekeeping ability, attention control, and other factors using a whole-task approach may be another useful avenue for future research.

## 6. Limitations

We note four limitations of this study. First, a psychometric network analysis of the individual observed variables (see Appendix B) suggested that some of the cognitive constructs we considered in this study are more coherent than others, at least as measured by the tasks we used in this study. Most notably, while there was a strong and statistically significant partial correlation between UNRAVEL and Letterwheel (*pr* = 0.35), only half of the links between the attention control variables (5 of 10) were significant. This suggests that the attention control tasks we used in this study may measure different aspects of attention control and/or other cognitive abilities. It is also possible that some of the links between attention control variables reflect method variance, especially the link between the two tasks that used a response deadline procedure (Stroop Deadline and Flanker Deadline, *pr* = 0.36). Note that some of the attention control tasks we used in this study have been refined and other attention control tasks have been introduced (see the “Squared” tasks of [11]). In future studies, we hope to examine whether results using these new attention control tasks are different than those reported here, especially with regard to predictive validity. Overall, these results suggest that network analysis could be a useful converging operation to evaluate the coherence of putative psychological constructs.

A second limitation is that we focused on only one aspect of job-relevant performance—multitasking—and measured that using the synthetic work approach. Multitasking is a requirement of many occupations in the military and civilian workplace ([41]), but the evidence presented here is no guarantee that Gf and placekeeping ability will significantly predict performance in any particular job as a whole (e.g., air-traffic control). Thus, it will be critical to evaluate whether the factors we tested predict performance in actual jobs above and beyond the ASVAB score. From our perspective, it would be ideal for studies to collect both measures of synthetic work performance and measures of real-world work performance.

A third limitation is the representativeness of the sample. A well-known problem with using “samples of convenience” in psychological research (viz., undergraduate students) is that the participants may not accurately reflect the population of interest. As already mentioned, the problem is acute in studies of cognitive ability because undergraduate students have typically already been selected on cognitive ability (i.e., score on a college admissions test). As indicated by college admissions test scores and comparison of ASVAB subtest means to those for a military sample, there was a relatively wide range of cognitive ability in our sample. However, there was some degree of range restriction in our sample (as reflected in higher means and smaller *SD*s for the sample on college admissions tests relative to all test takers). Thus, the effect sizes reported here may be underestimates of relationships that would be observed in the general population.

One approach to addressing this limitation might be online testing, which allows for larger and more representative samples to be collected than are practical to obtain with in-person studies. Online testing may also make it easier to test military personnel (or civilian employees) and would make it easier to obtain stratified samples to investigate questions such as whether racial/ethnic group differences are smaller in measures of executive functions than those found on standardized cognitive ability tests ([8]). The testing environment of in-person studies is obviously different than that of online studies, beginning with the presence of an experimenter in the former but not the latter. Therefore, it would be critical to validate online tasks before drawing conclusions from them.

Finally, it is possible that some broad, unmeasured cognitive factor explains the effects of both Gf and placekeeping ability on multitasking performance. Processing speed could be one such factor, although it is generally found to explain quite small amounts of inter-individual variability in complex task performance (e.g., [15]). A more likely possibility is working memory capacity (WMC). Historically, WMC has been closely associated with Gf (e.g., [32]; [28]), and may also be associated with placekeeping ability (but see [9]). Especially relevant, in a study of applicants for air-traffic control training, [14] ([14]) found that Gf added negligibly to the prediction of multitasking performance, as assessed with laboratory tasks, independent of WMC. As we noted earlier, we did collect a single measure of WMC (Mental Counters), and when we included it as a variable in our psychometric network analyses the results were largely unchanged and the conclusions were the same (see Appendix D). Moreover, whereas the bivariate correlations of WMC with AFQT (*r* = 0.38) and multitasking performance (*r* = 0.54) were significant, the corresponding partial correlations were nonsignificant (*pr*s = −0.06 and 0.05, respectively). Thus, the data we have suggests that WMC may not contribute to the relationship between AFQT and multitasking performance independent of Gf, placekeeping ability, and attention control. At the same time, this result may be specific to the Mental Counters measure, which does not always correlate highly with other measures of WMC (e.g., [27]). Replication with a composite variable based on multiple measures of WMC may be a useful avenue for future work.

## 7. Conclusions

Psychometric network analyses revealed that measures of two cognitive abilities—Gf and placekeeping ability—largely explained the relationship between AFQT score from the ASVAB and job-relevant multitasking performance. From a theoretical perspective, this evidence suggests that high levels of multitasking performance can be understood in terms of success in developing multitasking strategies (as measured by Gf) and success in implementing strategies during performance (as measured by placekeeping ability). From an applied perspective, our results suggest new directions for the development of cognitive test batteries that are even more predictive of job performance than those currently available. Finally, from a methodological perspective, the study illustrates the usefulness of psychometric network analysis in cognitive research. This approach could be extended to investigating the explanatory relationship between cognitive ability and societally relevant outcomes such as academic achievement, health, and mortality ([25]).

## Figures and Tables

**Figure 1 jintelligence-11-00225-f001:**
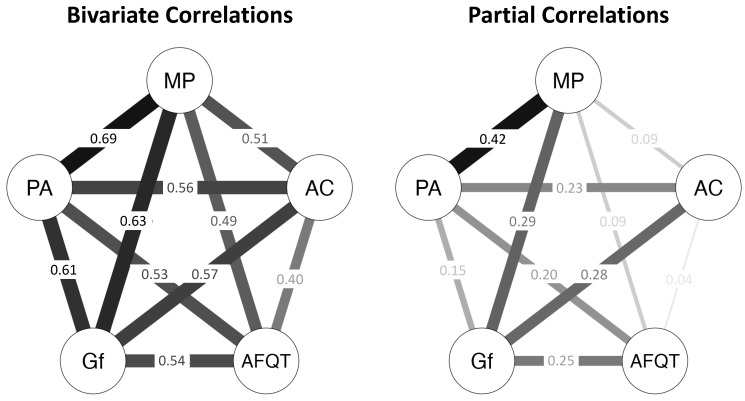
Psychometric network models with composites of observed variables. Values in the left panel are bivariate (Pearson’s) correlations (*r*s); values in the right panel are partial correlations (*pr*s). AFQT, Armed Services Qualification Test; Gf, Fluid Intelligence; PA, Placekeeping Ability; AC, Attention Control; MP, Multitasking Performance. *r*s > 0.11 and *pr*s > 0.12 are statistically significant (*p* < .05).

**Figure 2 jintelligence-11-00225-f002:**
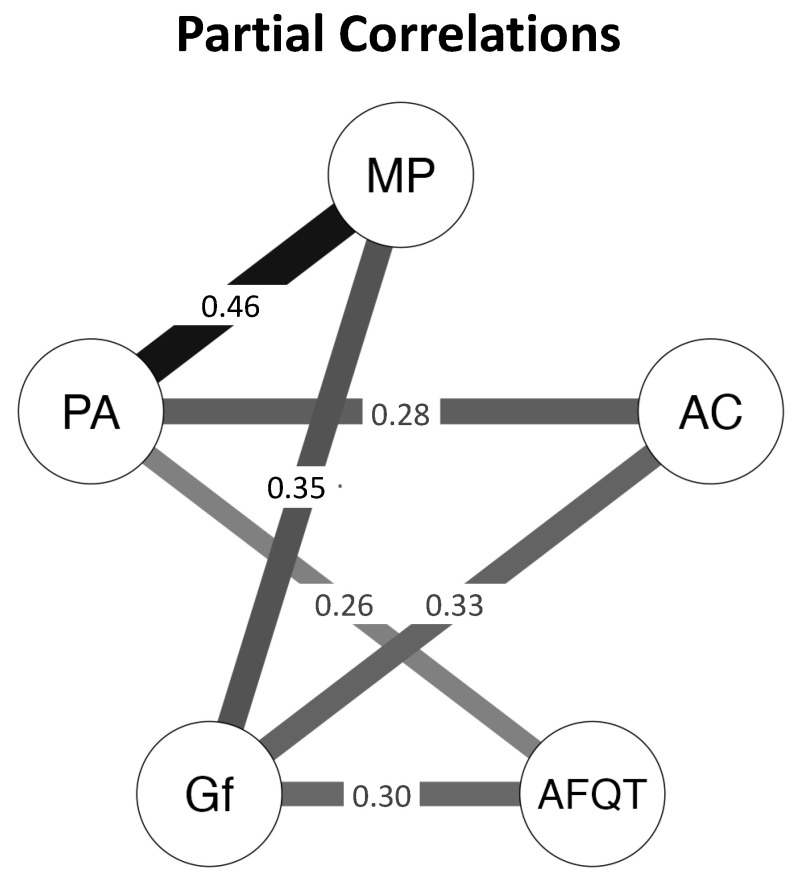
Bayesian psychometric network models with composites of observed variables. Values are partial correlations (*pr*s). Inclusion criterion, BF_10_ > 3. AFQT, Armed Services Qualification Test; Gf, Fluid Intelligence; PA, Placekeeping Ability; AC, Attention Control; MP, Multitasking Performance.

**Table 1 jintelligence-11-00225-t001:** Description of ASVAB Subtests.

Subtest	Acronym	Description	Items	Time
General Science	GS	Knowledge of physical and biological sciences	25	11
Arithmetic Reasoning *	AR	Ability to solve arithmetic word problems	30	36
Word Knowledge *	WK	Knowledge of and ability to infer word meanings	35	11
Paragraph Comprehension *	PC	Ability to obtain information from written passages	15	13
Mathematics Knowledge *	MK	Knowledge of high school-level mathematics concepts	25	24
Electronics Information	EI	Knowledge of electrical principles and electronic devices	20	9
Automotive and Shop Information	AS	Knowledge of automotive technology, shop terminology	25	11
Mechanical Comprehension	MC	Knowledge of mechanical and physical principles	25	19
Assembling Objects	AO	Ability to imagine how objects look when disassembled	25	15

Note. *, subtest used to compute the Armed Forces Qualification Test score, using the formula 2(WK + PC) + MK + AR. Subtest descriptions paraphrased from [55] ([55]).

**Table 2 jintelligence-11-00225-t002:** Assessments by Session in Order Administered.

Session 1	Session 2	Session 3	Session 4
Consent form Demographic form	Foster Task (MP)	Control Tower (MP)	SynWin (MP)
Arithmetic Reasoning (ASVAB)	UNRAVEL (PA)	Letter Sets (Gf)	Stroop Deadline (AC)
Word Knowledge (ASVAB)	Mental Counters (WMC)	Antisaccade (AC)	Selective Visual Arrays (AC)
Mathematics Knowledge (ASVAB)	Raven’s Matrices (Gf)	Letterwheel (PA)	
Paragraph Comprehension (ASVAB)	Sustained Attention to Cue (AC)	Number Series (Gf)	
		Flanker Deadline (AC)	

Note. ASVAB, Armed Services Vocational Aptitude Battery; MP, Multitasking Performance. PA, Placekeeping Ability; WMC, Working Memory Capacity; Gf, Fluid Intelligence; AC, Attention Control.

**Table 3 jintelligence-11-00225-t003:** Weight Matrices for Psychometric Network Models.

Composite Variable	1	2	3	4	5
1. AFQT	-	0.25 ** [0.14, 0.36]	0.04 [−0.10, 0.17]	0.20 ** [0.05, 0.34]	0.09 [−0.03, 0.22]
2. Gf	0.54 *** [0.45, 0.62]	-	0.28 *** [0.15, 0.40]	0.15 * [0.02, 0.28]	0.29 *** [0.18, 0.41]
3. Attention Control	0.40 *** [0.30, 0.50]	0.57 *** [0.48, 0.65]	-	0.23 ** [0.08, 0.38]	0.09 [−0.05, 0.24]
4. Placekeeping Ability	0.53 *** [0.44, 0.61]	0.61 *** [0.53, 0.68]	0.56 *** [0.47, 0.63]	-	0.42 *** [0.31, 0.53]
5. Multitasking Performance	0.49 *** [0.40, 0.58]	0.63 *** [0.56, 0.70]	0.51 *** [0.42, 0.60]	0.69 *** [0.62, 0.75]	-

Note. *N* = 270. Values below the diagonal are bivariate (Pearson’s) correlations (*r*s); values above the diagonal are partial correlations (*pr*s). Values in brackets are 95% confidence intervals [lower bound, upper bound], bootstrapped for *pr*s. AFQT, Armed Services Qualification Test; Gf, Fluid Intelligence. *, *p* < .05. **, *p* < .01. ***, *p* < .001.

**Table 4 jintelligence-11-00225-t004:** Bayes Factors for Partial Correlation Network Model.

Composite Variable	1	2	3	4	5
1. AFQT	-	0.00	16.67	0.00	8.33
2. Gf	∞	-	0.00	2.33	0.00
3. Attention Control	0.06	∞	-	0.00	6.67
4. Placekeeping Ability	∞	0.43	∞	-	0.00
5. Multitasking Performance	0.12	∞	0.15	∞	-

Note. *N* = 270. Values below the diagonal are BF_10_, the odds of the data given the alternative hypothesis (H_1_: *pr* > 0). Values above the diagonal are 1/BF_10_, the odds of the data given the null hypothesis (H_0_: *pr* = 0). AFQT, Armed Services Qualification Test; Gf, Fluid Intelligence.

**Table 5 jintelligence-11-00225-t005:** Hierarchical Regression Analyses Predicting Multitasking Performance (Model 1: Each Cognitive Factor Over AFQT).

	**Δ*R*^2^**	**Δ*F***	** *df* **	**β [95% CI]**	** *t* **
Model 1A					
Step 1	0.241	85.17 ***	1, 268		
AFQT				0.49 [0.39, 0.60]	9.23 ***
Step 2	0.193	91.06 ***	1, 267		
AFQT				0.21 [0.10, 0.32]	3.83 ***
Gf				0.52 [0.41, 0.63]	9.54 ***
Model 1B					
Step 1	0.241	85.17 ***	1, 268		
AFQT				0.49 [0.39, 0.60]	9.23 ***
Step 2	0.257	136.62 ***	1, 267		
AFQT				0.18 [0.08, 0.28]	3.45 **
PA				0.60 [0.50, 0.70]	11.69 ***
Model 1C					
Step 1	0.241	85.17 ***	1, 268		
AFQT				0.49 [0.39, 0.60]	9.23 ***
Step 2	0.119	49.54 ***	1, 267		
AFQT				0.34 [0.23, 0.44]	6.33 ***
AC				0.38 [0.27, 0.48]	7.04 ***

Note. *N* = 270. AFQT, Armed Services Qualification Test; Gf, Fluid Intelligence; PA, Placekeeping Ability; AC, Attention Control. Δ*R*^2^, change in variance explained. β, standardized regression coefficient. **, *p* < .01. ***, *p* < .001.

**Table 6 jintelligence-11-00225-t006:** Hierarchical Regression Analyses Predicting Multitasking Performance (Model 2: Each Cognitive Factor Over AFQT and Each Other Cognitive Factor).

	Δ*R*^2^	Δ*F*	*df*	β [95% CI]	*t*
Model 2A					
Step 1	0.498	132.44 ***	2, 267		
AFQT				0.18 [0.08, 0.28]	3.45 **
PA				0.60 [0.50, 0.70]	11.69 ***
Step 2	0.055	32.46 ***	1, 266		
AFQT				0.08 [−0.02, 0.18]	1.62
PA				0.46 [0.35, 0.56]	8.39 ***
Gf				0.31 [0.20, 0.42]	5.70 ***
Model 2B					
Step 1	0.360	75.06 ***	2, 267		
AFQT				0.34 [0.23, 0.44]	6.33 ***
AC				0.38 [0.27, 0.48]	7.04 ***
Step 2	0.100	49.42 ***	1, 266		
AFQT				0.18 [0.08, 0.29]	3.38 **
AC				0.20 [0.09, 0.31]	3.58 ***
Gf				0.42 [0.31, 0.54]	7.03 ***
Model 2C					
Step 1	0.434	102.42 ***	2, 267		
AFQT				0.21 [0.10, 0.32]	3.83 ***
Gf				0.52 [0.41, 0.63]	9.54 ***
Step 2	0.118	70.43 ***	1, 266		
AFQT				0.08 [−0.02, 0.18]	1.62
Gf				0.31 [0.20, 0.42]	5.70 ***
PA				0.46 [0.35, 0.56]	8.39 ***
Model 2D					
Step 1	0.360	75.06 ***	2, 267		
AFQT				0.34 [0.23, 0.44]	6.33 ***
AC				0.38 [0.27, 0.48]	7.04 ***
Step 2	0.156	85.87 ***	1, 266		
AFQT				0.15 [0.05, 0.25]	2.97 **
AC				0.16 [0.06, 0.27]	3.15 **
PA				0.52 [0.41, 0.63]	9.27 ***
Model 2E					
Step 1	0.434	102.42 ***	2, 267		
AFQT				0.21 [0.10, 0.32]	3.83 ***
Gf				0.52 [0.41, 0.63]	9.54 ***
Step 2	0.026	12.84 ***	1, 266		
AFQT				0.18 [0.08, 0.29]	3.38 **
Gf				0.42 [0.31, 0.54]	7.03 ***
AC				0.20 [0.09, 0.31]	3.58 ***
Model 2F					
Step 1	0.498	132.44 ***	2, 267		
AFQT				0.18 [0.08, 0.28]	3.45 **
PA				0.60 [0.50, 0.70]	11.69 ***
Step 2	0.018	9.95 ***	1, 266		
AFQT				0.15 [0.05, 0.25]	2.97 **
PA				0.52 [0.41, 0.63]	9.27 ***
AC				0.16 [0.06, 0.27]	3.15 **

Note. *N* = 270. AFQT, Armed Services Qualification Test; Gf, Fluid Intelligence; PA, Placekeeping Ability; AC, Attention Control. Δ*R*^2^, change in variance explained. β, standardized regression coefficient. **, *p* < .01. ***, *p* < .001.

**Table 7 jintelligence-11-00225-t007:** Hierarchical Regression Analyses Predicting Multitasking Performance (Model 3: Each Cognitive Factor Over AFQT and Both Other Cognitive Factors).

	Δ*R*^2^	Δ*F*	*df*	β [95% CI]	*t*
Model 3A					
Step 1	0.516	94.57 ***	3, 266		
AFQT				0.15 [0.05, 0.25]	2.97 **
PA				0.52 [0.41, 0.63]	9.27 ***
AC				0.16 [0.06, 0.27]	3.15 **
Step 2	0.040	24.17 ***	1, 265		
AFQT				0.08 [−0.02, 0.18]	1.54
PA				0.43 [0.32, 0.54]	7.59 ***
AC				0.08 [−0.02, 0.19]	1.54
Gf				0.28 [0.17, 0.40]	4.92 ***
Model 3B					
Step 1	0.460	75.59 ***	3, 266		
AFQT				0.18 [0.08, 0.29]	3.38 **
Gf				0.42 [0.31, 0.54]	7.03 ***
AC				0.20 [0.09, 0.31]	3.58 ***
Step 2	0.096	57.59 ***	1, 265		
AFQT				0.08 [−0.02, 0.18]	1.54
Gf				0.28 [0.17, 0.40]	4.92 ***
AC				0.08 [−0.02, 0.19]	1.54
PA				0.43 [0.32, 0.54]	7.59 ***
Model 3C					
Step 1	0.553	109.51 ***	3, 266		
AFQT				0.08 [−0.02, 0.18]	1.62
Gf				0.31 [0.20, 0.42]	5.70 ***
PA				0.46 [0.35, 0.56]	8.39 ***
Step 2	0.004	2.36	1, 265		
AFQT				0.08 [−0.02, 0.18]	1.54
Gf				0.28 [0.17, 0.40]	4.92 ***
PA				0.43 [0.32, 0.54]	7.59 ***
AC				0.08 [−0.02, 0.19]	1.54

Note. *N* = 270. AFQT, Armed Services Qualification Test; Gf, Fluid Intelligence; PA, Placekeeping Ability; AC, Attention Control. Δ*R*^2^, change in variance explained. β, standardized regression coefficient. **, *p* < .01. ***, *p* < .001.

## Data Availability

The data are publicly available on OSF.

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
