# Peer review of "Explaining the Validity of the ASVAB for Job-Relevant Multitasking Performance: The Role of Placekeeping Ability"

_jintelligence, 2023, doi:10.3390/jintelligence11120225_

Round 1

Reviewer 1 Report

Comments and Suggestions for Authors

Jintelligence_2510550

Explaining the Validity of the ASVAB for Job-Relevant Multitasking Performance: The Role of Placekeeping Ability

EVALUATION

Unfortunately, I cannot see the relevance of this report to increase our current knowledge regarding the ‘causes’ behind the predictive validity of the ASVAB.

Once upon a time there was a huge research project, funded by the US Air Force, at the Armstrong Laboratory, in which researchers tried to find out the cognitive mechanisms that might enhance the predictive validity of standardized ‘classic’ batteries such as the ASVAB. The research group, led by Patrick C Kyllonen, currently at the ETS, failed to substantiate the goal. They even developed an exhaustive battery based on the cognitive psychology approach (CAM Battery). It is really unfortunate that the authors of the present study choose to ignore this great research effort:

https://app.dimensions.ai/details/publication/pub.1043713973

Here are two examples of the observed findings when the classic psychometric and the cognitive approaches were compared:

https://psycnet.apa.org/record/1995-03689-001

https://www.sciencedirect.com/science/article/abs/pii/S016028962100101X

Therefore, the implication that the authors of this report rise here are highly arguable, to say the least.

The limitations noted by the authors themselves fail to exhaust the deep problems of their study, but perhaps the main one is the lack of ecologically valid criterion measures. The US Air Force research group named above tested the proper population with the disappointing result also noted above.

Incidentally, there is one study testing three hundred applicants for an air traffic control training course, and the reported results supported the starring role of working memory capacity and the null contribution of Gf (the key conclusion was substantiated by a re-analysis of an independent dataset) :

https://www.sciencedirect.com/science/article/abs/pii/S0160289610001078

I’m sorry, but my evaluation cannot be more positive. 

Author Response

Reviewer 1

Comment 1: Once upon a time there was a huge research project, funded by the US Air Force, at the Armstrong Laboratory, in which researchers tried to find out the cognitive mechanisms that might enhance the predictive validity of standardized ‘classic’ batteries such as the ASVAB. The research group, led by Patrick C Kyllonen, currently at the ETS, failed to substantiate the goal. They even developed an exhaustive battery based on the cognitive psychology approach (CAM Battery). It is really unfortunate that the authors of the present study choose to ignore this great research effort:

https://app.dimensions.ai/details/publication/pub.1043713973

Here are two examples of the observed findings when the classic psychometric and the cognitive approaches were compared:

https://psycnet.apa.org/record/1995-03689-001

https://www.sciencedirect.com/science/article/abs/pii/S016028962100101X

Therefore, the implication that the authors of this report rise here are highly arguable, to say the least.

Response: We agree that Kyllonen and colleagues’ research was important, which is why we cited it in the previous version of our manuscript. In this version we have added to our discussion of it (see p. 2, p. 5). However, we do not believe the Kyllonen work provides the last word on whether a “cognitive psychology approach” can help explain and improve the predictive validity of the ASVAB. Rather, we believe that recent work on cognitive control—ours included—can help explain the predictive validity of the ASVAB and inform the design of measures that may substantially improve its predictive validity.

We have also added a paragraph to our manuscript discussing Ree, Carretta, and Earles’ research (see p. 5). These authors found that variance specific to ASVAB subtests (“s” factors) adds negligibly to the prediction of job performance above and beyond variance common to the subtests (a “g” factor). Based on this finding, they argued that predicting job performance is “not much more than g.” However, the ASVAB does not include subtests for recently developed measures of cognitive control such as placekeeping, so whether such constructs can improve prediction of job-relevant performance is an open question.  

Comment 2: The limitations noted by the authors themselves fail to exhaust the deep problems of their study, but perhaps the main one is the lack of ecologically valid criterion measures. The US Air Force research group named above tested the proper population with the disappointing result also noted above.

Response: The reviewer’s position seems to be that the synthetic work approach we used to measure job-relevant performance has no value and that the only relevant criterion measure is the job itself. However, it is worth pointing out, as we now do in the manuscript (see p. 3), that the synthetic work approach was developed by the U.S. military and continues to be used in military-funded research. Furthermore, on logical grounds, we think that these two approaches to measuring job-relevant performance are informative and complementary. That is, whereas a study focusing on a particular job (e.g., air-traffic control) can establish the validity of a test for that job only, a synthetic work study can show that a test is likely to have predictive validity for a broad class of jobs (e.g., those requiring multitasking) and even potentially for jobs of the future. Thus, as we now note (see p. 20), it would be ideal for future studies to collect both types of criterion measures.

Comment 3: Incidentally, there is one study testing three hundred applicants for an air traffic control training course, and the reported results supported the starring role of working memory capacity and the null contribution of Gf (the key conclusion was substantiated by a re-analysis of an independent dataset) :

https://www.sciencedirect.com/science/article/abs/pii/S0160289610001078

Response: We thank the reviewer for pointing out this article (Colom et al., 2010) and now cite it in our manuscript (see p. 21). As we noted in the previous version of the manuscript, we did collect one measure of WMC, and when we included it in our psychometric network analyses, the results were unchanged (see Appendix D). That is, there were still independent and statistically significant contributions of Placekeeping Ability and Gf to Multitasking Performance. However, as we also pointed out, an important goal for future work is to collect additional measures of WMC. We will be surprised if the effects of Placekeeping Ability and Gf to Multitasking Performance go away, but it’s possible. All the more reason to follow up this study with one that includes an expanded set of predictor tasks. 

Reviewer 2 Report

Comments and Suggestions for Authors

Review of “Explaining the validity of the ASVAB for job-relevant multi-tasking performance:  The role of placekeeping ability”

Summary:  The authors present a large study examining variability in the AFQT.  Participants performed multiple measures of Gf, attention control, placekeeping, and multitasking. Network analysis suggested that although each composite was correlated with AFQT and multitasking, only Gf and placekeeping demonstrated significant partial correlations.  Thus, both Gf and placekeeping (but not attention control) accounted for unique variance in AFQT and multitasking.  Furthermore, Gf and placekeeping predicted unique variance in multitasking over and above that accounted for by AFQT and attention control.  The authors suggest that placekeeping and Gf are important for ASVAB performance and both can add incremental prediction to multitasking over ASVAB.

Evaluation:  This is an interesting paper that examines variation in ASVAB and multitasking. The writing is clear and the results are generally in line with the authors’ conclusions.  I only have a few minor questions/concerns.

First, I realize that the Engle group has suggested that the selective visual arrays task is a good measure of attention control, but I’m not convinced.  The reason is that the main indicator of performance is K across all trials which is not only measuring the ability to filter out distractors (filtering ability based on the work of Vogel and colleagues) but also just overall working memory capacity.  That is, you can have a low K in this task because you cannot filter out the distractors and your working memory is overloaded with them, or you can get a low score simply because you have low capacity to begin with.  That is, you might have perfect filtering abilities and block out the distractors, but you can only maintain 2-3 items in working memory resulting in lower task performance.  This is especially likely given that 5-7 target items are always presented.  Thus, there is no way of knowing what the low score represents.  Other work that has examined filtering abilities has included non-distractor trials and then used difference scores to try and isolate filtering from differences in capacity.  Although as you would expect, the reliability of these difference scores aren’t great.  Overall, I don’t think it is an issue for this paper, but I suggest not using the selective visual arrays task as a measure of attention control or at least trying to isolate filtering from capacity in future work.

Second, although I have no issues with the network analyses, I was a little surprised that these analyses and regressions were done instead of SEM.  Seems like SEM would be a perfect choice to examine some of the current questions.  Looking at the correlations among the composites, there are all pretty high.  So, an issue might be that with CFA and SEM the resulting latent correlations are so high that you get substantial multicollinearity.

Third, on p. 18 it is stated that Gf and placekeeping are causal factors in multitasking given that they account for unique variance.  However, it is possible that that variance is actually due to a third unmeasured variable.  Also, the fact that they account for unique variance and attention control does not, could be because they are simple broader factors. That is, in many cases Gf will account for unique variance given that it is much broader construct that lower level constructs.

Finally, on p. 18 it is speculated that differences between the current results and Martin et al. might be because Martin et al. sampled from GT and the community resulting in an extreme groups design.  But, their distributions are pretty normal and they get a high number of high ability participants from the community.  Thus, I don’t think there much evidence for this speculation.

Author Response

Reviewer 2

Comment 1: I realize that the Engle group has suggested that the selective visual arrays task is a good measure of attention control, but I’m not convinced.  The reason is that the main indicator of performance is K across all trials which is not only measuring the ability to filter out distractors (filtering ability based on the work of Vogel and colleagues) but also just overall working memory capacity.  That is, you can have a low K in this task because you cannot filter out the distractors and your working memory is overloaded with them, or you can get a low score simply because you have low capacity to begin with.  That is, you might have perfect filtering abilities and block out the distractors, but you can only maintain 2-3 items in working memory resulting in lower task performance.  This is especially likely given that 5-7 target items are always presented.  Thus, there is no way of knowing what the low score represents.  Other work that has examined filtering abilities has included non-distractor trials and then used difference scores to try and isolate filtering from differences in capacity.  Although as you would expect, the reliability of these difference scores aren’t great.  Overall, I don’t think it is an issue for this paper, but I suggest not using the selective visual arrays task as a measure of attention control or at least trying to isolate filtering from capacity in future work.

Response: We thank the reviewer for this point and will explore use of independent measures of filtering and capacity in future studies (although, as the reviewer notes, reliability of the difference scores is likely to be an issue). In this study we saw it as important to adopt the full suite of attention control tasks used in a previous study (Martin et al., 2020).

Comment 2: Second, although I have no issues with the network analyses, I was a little surprised that these analyses and regressions were done instead of SEM.  Seems like SEM would be a perfect choice to examine some of the current questions.  Looking at the correlations among the composites, there are all pretty high.  So, an issue might be that with CFA and SEM the resulting latent correlations are so high that you get substantial multicollinearity.

Response: This is a great point. We have now added discussion of the advantages of psychometric network analysis over SEM (see p. 4). The major advantage is that in psychometric network analysis there is no assumption of underlying latent factors (i.e., a cause common to a set of indicators). We have also added network analysis on the individual measures (see p. 11 and Appendix B, pp. 25-27). This analysis suggests that the placekeeping tasks capture a common set of cognitive operations that are independent of psychometric g, whereas the attention control tasks may capture distinct operations.

Comment 3: On p. 18 it is stated that Gf and placekeeping are causal factors in multitasking given that they account for unique variance. However, it is possible that that variance is actually due to a third unmeasured variable. Also, the fact that they account for unique variance and attention control does not, could be because they are simple broader factors. That is, in many cases Gf will account for unique variance given that it is much broader construct that lower-level constructs.

Response: We agree and have added this point (see p. 21). One possible third variable is working memory capacity (WMC), although when we entered our measure of WMC into the analysis, the linkages of Placekeeping Ability and Gf to Multitasking Performance were essentially unchanged. Another possibility is processing speed. However, we would be surprised if this is the case. Although processing speed is typically found to account for large amounts of age-related variance in criterion task performance (e.g., multitasking performance, see Salthouse et al., 1995), it usually does not explain significant amounts of between-subjects variance (e.g., Conway et al., 2002, now cited on p. 21).

Comment 4: Finally, on p. 18 it is speculated that differences between the current results and Martin et al. might be because Martin et al. sampled from GT and the community resulting in an extreme groups design. But, their distributions are pretty normal and they get a high number of high ability participants from the community. Thus, I don’t think there much evidence for this speculation.

Response: Thanks for pointing this out. We have removed this speculation (see p. 19) and have added the point that the difference across the two studies may simply be due to sampling error.

Reviewer 3 Report

Comments and Suggestions for Authors

The present study examines individual differences in placekeeping ability, fluid intelligence, and attention control as potential mediating variables in explaining the association between the Armed Forces Qualifying Test (AFQT) and multitasking performance. To that end, they had a sample of undergraduate students complete a battery of attention, multitasking, and fluid intelligence tasks in addition to the AFQT. Then, using network models, they assessed the bivariate and partial correlations between the factors. Factor scores were estimated using a (sometimes weighted) average of standardized scores. They found that fluid intelligence and placekeeping ability both partially explained the AFQT-multitasking relation, and each explained a portion of unique variance in multitasking, whereas attention control did not.

Overall, my evaluation of the manuscript was positive. I believe this manuscript should eventually be accepted for publication. However, I do have a few questions and comments I would like to see addressed.

First, for the network model, was the spatial positioning of the nodes constrained? I am not an expert in network modeling, but I am used to seeing the positioning of the nodes be freely estimated, so the spatial positioning of the nodes provides additional information regarding the relations in addition to the connection strength.

Second, was the network analysis ever performed on the task-level data? This felt like a skipped step. For example, in structural equation modeling with latent variables, a measurement model (confirmatory factor analysis) is often specified first. This allows a demonstration that the putative measures of each latent construct do indeed load onto those constructs in a theoretically coherent manner. Then, the latent variables are used in regression analyses. It would behoove this paper to have a similar demonstration showing similar relatedness among the measures of each putative construct before moving to the factor-level analyses. Again, I wonder if allowing the spatial positioning of the nodes to be freely estimated would provide some additional information.

Third, in general, I like network modeling because it is a nice visualization of bivariate and partial correlations. However, in this case, what is the advantage of using it over “traditional” methods like confirmatory/exploratory factor analysis, beyond just visualization? Indeed there may be many, but those could be explained early in the method or at the end of the introduction.

As a minor note, the OSF link provided is view only, and it did not seem like this was intentional. Please consider making the page public so the analyses can be independently vetted before publication.

Author Response

Reviewer 3

Comment 1: Overall, my evaluation of the manuscript was positive. I believe this manuscript should eventually be accepted for publication. However, I do have a few questions and comments I would like to see addressed.

Response: We thank Reviewer 3 for their time in reviewing our manuscript and for the constructive comments.      

Comment 2: [F]or the network model, was the spatial positioning of the nodes constrained? I am not an expert in network modeling, but I am used to seeing the positioning of the nodes be freely estimated, so the spatial positioning of the nodes provides additional information regarding the relations in addition to the connection strength.

Response: It is true that freely estimated positioning would convey additional information about the strength of relations (beyond link width and color). In fact, in our newly added Figure B1, position is freely estimated. However, we would like to keep the main figures as they are because the symmetrical positioning makes them easier for the reader to interpret.

Comment 3: [W]as the network analysis ever performed on the task-level data? This felt like a skipped step. For example, in structural equation modeling with latent variables, a measurement model (confirmatory factor analysis) is often specified first. This allows a demonstration that the putative measures of each latent construct do indeed load onto those constructs in a theoretically coherent manner. Then, the latent variables are used in regression analyses. It would behoove this paper to have a similar demonstration showing similar relatedness among the measures of each putative construct before moving to the factor-level analyses. Again, I wonder if allowing the spatial positioning of the nodes to be freely estimated would provide some additional information.

Response: We appreciate this suggestion and have added a task-level network analysis (p. 11 and Appendix B, pp. 25-27). The results are mixed in that, for factors defined in the extant literature, the component variables are only somewhat related. In particular, attention control seems to measure multiple abilities. In contrast, for the placekeeping factor, which is the relatively novel one that we bring to the table, the component variables were related. The results suggest that network analysis may be a helpful converging operation for evaluating the coherence of putative constructs. For present purposes, we form composites of observed variables for attention control, Gf, AFQT, and multitasking based on previous research that has found, or assumed, that they are coherent constructs. We think this is the best way to relate our findings to previous work—but also that this task-level analysis is an important addition in that it suggests that assumptions made in previous work may need to be revisited. In Appendix B, we have also added a psychometric network analysis using the two Gf measures that cohere (Raven’s, Letter Sets) and the three attention control measures that cohere the most (Stroop Deadline, Flanker Deadline, Antisaccade). The results are very similar to those in the main analysis (Figure 1) and conclusions are unchanged (see Figure B2, p. 27).

Comment 4: [I]n general, I like network modeling because it is a nice visualization of bivariate and partial correlations. However, in this case, what is the advantage of using it over “traditional” methods like confirmatory/exploratory factor analysis, beyond just visualization? Indeed there may be many, but those could be explained early in the method or at the end of the introduction.

Response: This point echoes one from Reviewer 2. We have added a discussion of the advantages of psychometric network analysis on page 4.

Comment 5: As a minor note, the OSF link provided is view only, and it did not seem like this was intentional. Please consider making the page public so the analyses can be independently vetted before publication.

            Response: Thank you for catching this. Problem fixed.

Round 2

Reviewer 1 Report

Comments and Suggestions for Authors

Thanks for your revision of the Ms.

Reviewer 2 Report

Comments and Suggestions for Authors

Review of “Explaining the validity of the ASVAB for job-relevant multi-tasking performance:  The role of placekeeping ability”

Summary:  The authors present a large study examining variability in the AFQT.  Participants performed multiple measures of Gf, attention control, placekeeping, and multitasking. Network analysis suggested that although each composite was correlated with AFQT and multitasking, only Gf and placekeeping demonstrated significant partial correlations.  Thus, both Gf and placekeeping (but not attention control) accounted for unique variance in AFQT and multitasking.  Furthermore, Gf and placekeeping predicted unique variance in multitasking over and above that accounted for by AFQT and attention control.  The authors suggest that placekeeping and Gf are important for ASVAB performance and both can add incremental prediction to multitasking over ASVAB.

Evaluation:  The authors have addressed all of my concerns.  I recommend acceptance.

Reviewer 3 Report

Comments and Suggestions for Authors

I am happy to accept, but I noticed that the Foster Task (FT) is not defined in the caption of Figure B1. Please check other captions, too.